# Beyond Autoregression: Permutation Invariant Graph Generation with Scalable Edge Construction

## Abstract

Graph generation models have advanced significantly with deep learning, yet they remain limited in scalability, flexibility, and ability to model underlying structures. We present *GraphK*, a novel encoder-sampler-decoder framework for graph generation that overcomes these challenges through structural flexibility and computational efficiency. Unlike autoregressive approaches constrained by vocabulary size (i.e. number of nodes in graph generation), GraphK allows for both upscaling (generating graphs with more nodes than the input) and downscaling, providing a flexible control over output graph size. By learning permutation-invariant latent representations and sampling new node embeddings via maximum likelihood estimation, GraphK generalizes across graph sizes and structures. For edge generation, we employ edge prediction with a KDTree-based top-$k$ neighbor search in the latent space, reducing computational cost. Based on the manifold smoothness assumption, our method effectively captures graph properties. Experiments on synthetic and real-world datasets show that GraphK outperforms existing methods, accurately learns graph structures, and generates synthetic graphs without explicit definitions.

## 1 Introduction

Graph generation plays a crucial role in modeling and analyzing complex relationships among entities, with wide-ranging applications in domains such as software engineering Brockschmidt et al. (2018); Allamanis et al. (2017), recommender systems Wang et al. (2021), and chemistry Liu et al. (2018). As the diversity of graph-structured data and applications expands, the ability to generate synthetic yet realistic graphs has become increasingly valuable. Graph generation involves the creation of new graphs that preserve specific structural properties of observed real-world graphs, enabling tasks such as data augmentation (Abul'atta et al., 2021; Bas et al., 2024; Bojchevski et al., 2018), simulation (Leskovec et al., 2010), and code completion (Brockschmidt et al., 2018). The study of graph generation has a long history rooted in probabilistic modeling. Classical models such as the Erdős–Rényi random graph, the Barabási–Albert model, and the stochastic block model (SBM) have provided foundational insights into the structural properties of real-world graphs, including sparsity and scale-free distributions (Erdos & Renyi, 1959; Barabasi & Albert, 1999; Holland et al., 1983). While these models are simple, they are often limited in their expressive power, failing to capture higher-order structural dependencies or attribute correlations in complex graph data. More importantly, these methods rely on predefined assumptions (e.g., number of nodes, edge probabilities) about graph structure rather than learning these directly from data.

In contrast, recent advances in deep learning-based graph generation seek to overcome these limitations by learning semantic and structural patterns directly from graph data. Notable among these are deep generative models, including variational autoencoder (VAE)-based methods (e.g., GraphVAESimonovsky & Komodakis (2018), GraphiteGrover et al. (2018), JT-VAEJin et al. (2018)), autoregressive frameworks (e.g., GraphRNNYou et al. (2018), GRANLiao et al. (2020)), and diffusion models (e.g., DiGress, score-based models, spectral diffusion) Vignac et al. (2022); Kong et al. (2023); Wen et al. (2024). Additionally, flow-based models Liu et al. (2023), though relatively niche, form a significant part of this landscape. Since these models represent a "probability-based" deep learning approach and explicitly model the underlying probability distributions, unlike the implicit

generation mechanisms of autoregressive models such as GraphRNN or the approximate posterior inference used in VAEs. While the strengths and weaknesses of these models are discussed in detail in Sec. 2, we briefly highlight the key limitations: (1) sensitivity to node ordering (i.e., lack of permutation invariance), (2) difficulty scaling beyond training graph sizes, and (3) high computational demands. Addressing these limitations simultaneously remains an open problem, which motivates the development of our proposed framework, **GraphK**, a novel graph generation approach that balances structural flexibility, permutation invariance, and computational efficiency. Inspired by the encoder-sampler-decoder paradigm Faez et al. (2020), our approach leverages structural embeddings, Gaussian Mixture Models (GMM), and efficient geometric decoding to overcome critical limitations found in prior deep generative graph methods. In what follows, we detail how GraphK addresses each of these limitations.

**(1) Lack of permutation invariance:** Permutation invariance refers to the property that the generation process should not depend on how nodes are labeled or ordered in the input. This is crucial, as isomorphic graphs can have multiple valid representations, and a robust generative model should treat them equivalently. Autoregressive models like GraphRNNYou et al. (2018) depend heavily on specific node sequencing, and models like GraphVAESimonovsky & Komodakis (2018) require expensive post-hoc matching processes to mitigate order biases. GRAN attempted to alleviate ordering bias by averaging multiple orderings but could not eliminate it entirely. In contrast, GraphK achieves full permutation invariance by employing structurally informative embeddings (e.g., Node2Vec Grover & Leskovec (2016) or VGAEKipf & Welling (2016)) as its encoding step, and GMM in the sampling step.

**(2) Limited scalability in node count:** Most deep generative models are trained on a certain size distribution and have limited ability to scale flexibly beyond the node counts observed during training. GraphK overcomes this limitation through a latent-space sampling strategy that fully decouples the graph size from the training data. By fitting a GMM to the node embeddings, our framework enables flexible sampling of any desired number of latent points, allowing the generation of larger or smaller graphs while preserving key structural patterns from the original input. To the best of our knowledge, this is the first method to explicitly incorporate a principled upscaling mechanism into the graph generation process, offering a practical capability not addressed in prior work.

**(3) High computational cost:** Training and inference in prevalent deep generative architectures, particularly diffusion models and autoregressive methods, often demand substantial computational resources. They typically exhibit quadratic or higher complexity with increasing graph size. To overcome this, GraphK adopts a highly efficient geometric decoder based on a k-nearest neighbor (kNN) strategy. Instead of performing an exhaustive $O(N^2)$ search over node pairs to determine connectivity, our method constructs a KD-tree Bentley (1975) on sampled latent embeddings, enabling fast nearest neighbor queries with complexity $O(N \log N)$.

In summary, our proposed method, GraphK, provides an effective solution in the field of deep graph generation by capturing structural patterns and being robust to node permutations. By combining structural embeddings that ensure permutation invariance, latent-space sampling for scalable size-flexible graph generation, and efficient geometric decoding via nearest-neighbor searches, our approach offers a computationally lightweight solution. These strengths make it well-suited for a wide range of applications, including social, infrastructure, biological, and knowledge networks, with particular effectiveness in generating community-rich graphs, where latent structure and modularity are especially prominent.

## 2 RELATED WORK

We categorize and review related graph generation methods in three main groups. Table 1 also provides a summary comparison, offering a quick overview of their capabilities and the limitations discussed in Sec. 1. While we acknowledge the broader landscape of advancements in graph generation, the selected methods have been carefully chosen based on their relevance to the approach.

**Variational Autoencoder (VAE) Approaches.** Classical models are fast and able to generate very large graphs; however, they assume node identities are fixed or irrelevant (graphs are typically generated on labeled nodes or exchangeable node slots), which means they do not naturally learn features from data. These limitations set the stage for data-driven deep models, where variational autoen-

coders were among the first deep learning methods applied to graph generation. Here, GraphVAE Simonovsky & Komodakis (2018) is a seminal work that encodes an input graph using a graph neural network, obtaining a latent vector as in GraphK, and then decodes an entire adjacency matrix in one shot. A major limitation of early VAEs like GraphVAE is their poor scalability in graph size. The decoder must output an adjacency matrix of size $N_{\max}^2$, which was applied to graphs with at most 20-30 nodes. Moreover, GraphVAE's independent edge assumption and small latent dimension made it rely heavily on local structures. It can capture local motifs that it saw during training, but it misses higher-order patterns like communities or long-range dependencies. Subsequent VAE-based methods addressed some of these limitations. Graphite Grover et al. (2018) showed improved results on larger graphs (up to hundreds of nodes) by avoiding the explicit graph matching step; and using permutation-equivariant networks, as also seen in another variant J-Tree VAE Jin et al. (2018). However, their main weakness in modeling complex dependencies (due to the often factorized decoder) led to the development of autoregressive models, which tackle these dependencies by generating graphs step by step.

**Deep Autoregressive Models** Autoregressive models generate graphs one element at a time, node by node or edge by edge, always conditioning on what has been generated so far. GraphRNN You et al. (2018) treats graph generation as a sequence generation problem using two levels of RNN; DeepGMG Li et al. (2018) also generates graphs by adding nodes and edges sequentially, but instead of using two RNNs and a fixed BFS order, it uses a GNN at each step to make decisions. Both proved capable of learning distributions far more complex than most studies, but their computational complexities due to their sequential nature still limit their application to graphs with thousands of nodes. To address this, GRAN Liao et al. (2020) uses the idea of generating graphs in blocks of nodes with an attention mechanism to capture dependencies while relaxing the strict ordering constraints imposed by RNN-based models. It provided a path to generate much larger graphs than previously possible, but still, the marginalization over orderings and the large GNNs add overhead for the training. More importantly, GRAN is not fully permutation-invariant because it cannot average over all $n!$ orders (i.e., intractable); simply, by sampling a diverse set of orderings, it tries to approximate invariance.

**Denoising Diffusion Models.** One of the most exciting recent developments in generative modeling is the rise of denoising diffusion models, also known as score-based models. Two pioneering works in this area for graphs are EDP-GNN Kong et al. (2023), which was the first to bring score-based generative modeling to graphs, and GDSS Wen et al. (2024), which extended diffusion to a continuous-time framework. More recently, DiGress Vignac et al. (2022) implements a discrete diffusion on graphs via edge-level noise and treats graph learning as the discretization of a partial differential equation (PDE). This enables smooth and stable transitions during generation, making it especially useful for dynamic or evolving graph structures. However, the output of graph size in the generation is still moderate (molecules and planar graphs in benchmarks are at most a few dozen nodes), and it does not explicitly demonstrate the generation of huge single graphs.

Table 1: Overview of related works and their limitations.

| Method | Core Approach | Focus | Node Order Invariance | Computational Complexity (per graph) | Up-scaling |
|---|---|---|---|---|---|
| ER Erdos & Renyi (1959) | Random graph generation based on edge probability | General | No | Low. $\Rightarrow$ Generates edges by one Bernoulli draw per pair | No |
| BA Barabasi & Albert (1999) | Preferential attachment | Scale-free networks | No | Low. $\Rightarrow$ Adds one node with $\mathcal{O}(1)$ preferential links | No |
| SBM Holland et al. (1983) | Community structure with probabilistic edge connections | Community | No | Low. $\Rightarrow$ Samples block-wise probabilities. | No |
| GraphRNN You et al. (2018) | Autoregressive node and edge sequence generation | General | No | High. $\Rightarrow$ Sequential edge writing: worst-case $\mathcal{O}(N^2)$ decisions; BFS ordering shortens average run-time but still quadratic on dense graphs. Up to $\approx$ 80 nodes. | No |
| GRAN Liao et al. (2020) | Recurrent attention over graph structure | General | Partial | Low. $\Rightarrow$ Adds $b-node$ blocks; $\approx \lceil N/b \rceil$ GNN passes. With small constant $b$ the cost grows roughly $\mathcal{O}(N)$. Shown to $\approx$ 5000 nodes. | No |
| Digress Vignac et al. (2022) | Diffusion denoising with score-based model (transformer denoiser) | General | Yes | Low. $\Rightarrow$ Randomly add/remove edges and noise attributes, then model learns to reverse this. | No |
| EDP-GNN Kong et al. (2023) | Score-based generative model (discrete Langevin diffusion) | General | Yes | High. $\Rightarrow$ Requires many sampling steps (hundreds of GNN passes); practical for graphs $\leq \mathcal{O}(100)$nodes. | No |
| GDSS Wen et al. (2024) | Score matching in latent and data space. Continuous-time diffusion | General | Yes | High. $\Rightarrow$ Fewer steps than discrete, but still requires many neural function evaluations; typically applied to graphs up to tens of nodes (e.g. molecules) | No |
| GraphVAE Simonovsky & Komodakis (2018) | Variational Autoencoder with edge-centric decoding | Small molecular graphs | Partial | High. $\Rightarrow$ Decoder $\mathcal{O}(N^2)$ + graph-matching $\approx \mathcal{O}(N^4)$ infeasible beyond $\approx$ 30 nodes. | No |
| J-TreeVAE Jin et al. (2018) | Hierarchical VAE with chemical substructures | Molecular graphs | Yes | High. $\Rightarrow$ Chemical assembly rules keep graphs $\leq$ 50 atoms. | No |
| Graphite Grover et al. (2018) | Latent graph refinement via iterative inference | General | Yes | Moderate. $\Rightarrow$ Each $\mathcal{O}(N+E)$; good on sparse graphs. Shown up to $\approx$ 300 nodes. | No |
| GraphK (Ours) | Permutation-invariant encoding, sampling via GMM, edge construction via KDTree | Social + Community | Yes | Low. $\Rightarrow$ KD-Tree build+kNN links $\mathcal{O}(NlogN)$. Shown to $\approx$ 10000 nodes, practically increase further. | Yes |

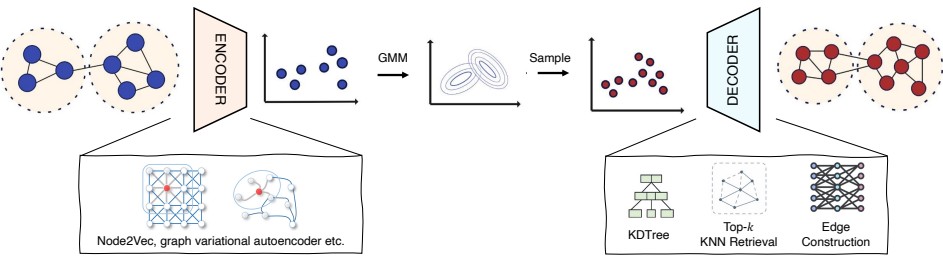

Figure 1: Overview of the GraphK architecture.

## 3 METHODOLOGY

In this section, we describe our proposed method, GraphK[1], a graph generation framework that follows an encoder–sampler–decoder architecture as shown in Fig. 1. The model is designed to learn the underlying graph structure from observed graphs and to generate new graphs that preserve this characteristic. Moreover, it supports flexible control over the number of nodes and edges in the generated graph, independent of the size of the input graph. GraphK consists of three main components: (i) the *encoder*, which maps input graphs into a latent space using flexible graph embedding techniques; (ii) the *sampler*, which generates new node embeddings by sampling from a learned latent distribution; and (iii) the *decoder*, which predicts graph connectivity among the sampled nodes using nearest neighbor-based edge construction mechanism. In the following, we provide detailed explanations of each component.

### ENCODER

The encoder module maps input graphs into a latent representation space that captures their structural and semantic properties. This module is designed to be modular and flexible, allowing the use of various graph embedding techniques. In our implementation, we used both shallow embedding methods, such as Node2Vec Grover & Leskovec (2016), and neural network-based encoders, such as the encoder component of a variational graph autoencoder (VGAE) Kipf & Welling (2016). However, the framework is compatible with any encoding method, making it adaptable to different tasks and data characteristics.

**General Formulation.** Let $G = (V, E)$ denote the input graph, where $V = \{v_1, v_2, ..., v_N\}$ is the set of nodes, $E$ is the set of edges. The encoder maps the graph into a set of latent embeddings $\mathcal{Z} = \{z_1, z_2, ..., z_N\}$, where each $z_i \in \mathbb{R}^d$ is the latent representation of node $v_i \in V$. This mapping is represented as:

$$z_i = f_{\text{enc}}(v_i; G), \quad \forall v_i \in V \tag{1}$$

**Node2Vec Encoder.** We use structural embedding methods to map nodes into a latent space. In practice, we experimented with both shallow embedding (e.g., Node2Vec Grover & Leskovec (2016)) and neural network-based encoders (e.g., VGAE Kipf & Welling (2016)). Full details of the Node2Vec formulation are provided in Appendix A.2.

**VGAE Encoder.** Alternatively, we used a neural encoder such as VGAE Kipf & Welling (2016), which encodes both the node features and graph structure into a probabilistic latent space. The VGAE encoder uses a Graph Convolutional Network (GCN) to parameterize the posterior distribution $q(z_i \mid X, A)$ as a multivariate Gaussian:

$$q(z_i \mid X, A) = \mathcal{N}(z_i \mid \mu_i, \text{diag}(\sigma_i^2)) \tag{2}$$

where $A \in \{0, 1\}^{|V| \times |V|}$ is the adjacency matrix, and the mean and variance vectors are computed as:

$$\mu = \text{GCN}_\mu(X, A), \quad \log \sigma = \text{GCN}_\sigma(X, A) \tag{3}$$

---

[1]All data and code will be publicly released upon acceptance.

Latent vectors $z_i$ are then sampled using the reparameterization trick:

$$z_i = \mu_i + \sigma_i \odot \epsilon, \quad \epsilon \sim \mathcal{N}(0, I) \tag{4}$$

This flexible design enables GraphK to leverage different types of encoders depending on the complexity of the input data and the desired trade-off between expressiveness and computational cost. Additionally, Equation 3 allows for the incorporation of node features, enabling richer representations when such information is available.

### SAMPLER

In the GraphK framework, the sampler module is responsible for generating new node embeddings by drawing samples from a learned distribution over the latent space. To capture the complex structure of this latent space, we employ a Gaussian Mixture Model (GMM) that assumes the latent representations are drawn from a mixture of several multivariate Gaussian distributions with unknown parameters. The GMM is parameterized by a set of mixture weights, mean vectors, and covariance matrices, which are estimated via Maximum Likelihood Estimation (MLE) using the latent embeddings produced by the encoder. This formulation enables the sampler to model diverse node representations that reflect the statistical properties of the original graph embeddings. Crucially, using a continuous and parameterized distribution allows the model to generate an arbitrary number of synthetic node embeddings, thereby supporting flexible graph generation that is not constrained by the number of nodes in the input graph.

Let the encoded node representations from the encoder be denoted as $\mathcal{Z} = \{z_1, z_2, ..., z_N\}$, where $z_i \in \mathbb{R}^d$ is a $d$-dimensional latent vector. We assume that each $z_i$ is generated from a mixture of $K$ Gaussian components (see implementation details in Appendix). The GMM models the probability density function as:

$$p(z_i) = \sum_{k=1}^{K} \pi_k \mathcal{N}(z_i \mid \mu_k, \Sigma_k) \tag{5}$$

where $\pi_k$ is the mixing coefficient for component $k$ such that $\sum_{k=1}^{K} \pi_k = 1$, and $\mathcal{N}(z_i \mid \mu_k, \Sigma_k)$ is a multivariate Gaussian distribution with mean $\mu_k \in \mathbb{R}^d$ and covariance matrix $\Sigma_k \in \mathbb{R}^{d \times d}$.

To estimate the parameters of the GMM, we employ the Expectation–Maximization (EM) algorithm, a standard procedure for latent-variable models. For completeness, the full update rules are provided in Appendix A.3.

### DECODER

The decoder reconstructs the graph structure by predicting edges between the sampled node embeddings. Specifically, it performs edge prediction that combines distance-based metrics (e.g., dot product or cosine similarity). For each potential node pair $(z_i, z_j)$, the decoder estimates a connection probability $p_{ij}$, which is used to form the adjacency matrix of the generated graph. This strategy helps avoid forming unrealistic or overly dense connections during decoding. Specifically, edges are reconstructed between nodes based on their latent representations. A naive approach would involve computing pairwise similarities for all possible node pairs, resulting in a computational complexity of $O(n^2)$, where $n$ is the number of nodes. To overcome this scalability bottleneck, we adopt a more efficient edge generation strategy based on the KDTree algorithm Bentley (1975).

**Latent Space Smoothness Assumption.** We assume that the learned latent space preserves a smooth manifold structure such that nodes with similar embeddings are more likely to share an edge. Formally, let $z_v \in \mathbb{R}^d$ denote the latent representation of node $v$. Under the smoothness assumption Dong et al. (2016), if the Euclidean distance $\|z_v - z_u\|_2$ is small, then nodes $v$ and $u$ are likely to be connected in the graph.

**Top-$k$ Nearest Neighbor Retrieval.** Let $\mathcal{V}$ be the set of all nodes with latent embeddings $Z = \{z_v \mid v \in \mathcal{V}\}$. For each node $v \in \mathcal{V}$, we aim to find its top-$k$ nearest neighbors based on Euclidean distance in the latent space. The neighborhood set $\mathsf{N}_k(v)$ is defined as:

$$\mathsf{N}_k(v) = \mathrm{argmin}^{(k)}_{u \in \mathcal{V} \setminus \{v\}} \|z_v - z_u\|_2 \qquad (6)$$

Here, $\mathrm{argmin}^{(k)}$ returns the $k$ nodes with the smallest distances to $z_v$. The parameter $k$ is selected based on the mode of the degree distribution of the input graphs, which provides a statistically grounded estimate of connectivity. Notably, the maximum degree of a node is not limited to $k$, allowing GraphK to generate power-law graphs (See Appendix A.9).

**Efficient Search with KDTree.** To compute $\mathsf{N}_k(v)$ efficiently, we construct a KDTree data structure over the set of latent embeddings $Z$. The KDTree partitions the latent space using axis-aligned hyperplanes and enables efficient nearest neighbor queries. The construction of the KDTree has average complexity $O(n \log n)$, and each query for top-$k$ neighbors can be performed in $O(\log n + k)$ time in low-dimensional spaces.

**Edge Construction.** Once $\mathsf{N}_k(v)$ is obtained for each node $v$, we add edges from $v$ to all nodes in $\mathcal{N}_k(v)$, effectively forming a locally connected graph structure. Optionally, these candidate edges can be further scored using a learnable edge decoder $f_\theta(z_v, z_u)$ to predict edge probabilities or weights.

This approach enables scalable and geometry-aware edge construction that is consistent with the statistical properties of the original graph, while significantly reducing the computational burden compared to exhaustive pairwise evaluation. Additionally, in graphs with well-defined community structures, the encoder may learn an embedding space where nodes from the same community are tightly clustered, potentially leading to a loss of inter-community edge information. To mitigate this, training a learnable edge decoder (i.e., edge prediction model based on neural networks) is beneficial. We also observed that during the training of the decoder, fine-tuning the learned embeddings with a small learning rate further improves performance.

**Exchangeability, projectivity, and sparsity.** A central tension in generative graph modeling is that permutation-invariant, size-consistent (projective) models are characterized by conditionally independent edges given latent variables (i.e., a graphon or latent-position form), via the Aldous-Hoover theorem and its generalizations. This representation is both sufficient and necessary for a broad class of projective models, including stochastic block models and VAE-style latent factor models (Orbanz & Roy, 2015; Xu et al., 2020). A well-known corollary is density: if the graphon's mean edge rate is non-zero, the expected number of edges scales as $\Theta(n^2)$, so jointly exchangeable graphs are almost surely dense (or empty) (Orbanz & Roy, 2015). Remedies that simply down-scale edge probabilities (e.g., $w/n$) enforce sparsity but do not yield realistic higher-order structure such as power-law degrees. Hamilton (2020) likewise frames the core challenge in graph generation as producing *realistic* structures, not just any exchangeable samples, while retaining tractability. In this paper, we take a pragmatic route. GraphK is permutation-invariant at the *embedding* level but deliberately departs from the fully exchangeable regime by enforcing locality via $k$-NN in latent space, which caps *proposed* edges at $O(kn)$ and thus yields linear-edge sparsity. Importantly, because $k$-NN selection is asymmetric, many nodes may choose the same target, allowing hub formation and heavy-tailed degrees despite each node emitting at most $k$ edges. Conceptually, GraphK addresses the open problem articulated by Orbanz & Roy (2015), reconciling symmetry principles with sparse network properties by permitting *limited* dependencies between edges, and advances a simple, scalable mechanism that empirically produces realistic community structure while keeping the model permutation-agnostic in its latent parametrization. Our framework thus also contributes a step toward aligning permutation-invariance with sparse graph generation, showing that even a lightweight algorithm like GraphK can produce realistic behavior. We further illustrate this effect in Appendix A.9, where experiments on the Gnutella05 Leskovec & Krevl (2014) network confirm that asymmetric $k$-NN selection in GraphK can reproduce power-law-like degree distributions.

## 4 Experimental Study

We tested GraphK by evaluating its ability to capture structural properties of both real-world and synthetic networks. Details on data preprocessing steps provided in Appendix A.4.1.

## 4.1 DATASETS

We evaluated DyGG on both synthetic and real-world temporal datasets with varying sizes and graph (e.g., protein, community and citation).

**Protein dataset.** A total of 918 protein graphs from Dobson & Doig (2003) are used, each containing between 100 and 500 nodes. In these graphs, each node represents an amino acid, and an edge is formed between two nodes if the distance between the corresponding amino acids is less than 6 Angstroms. We sampled 225 graphs from this dataset for use with GraphRNN as in their code settings.

**CiteSeer dataset.** The CiteSeer (CS) network Sen et al. (2008) is a widely used benchmark dataset where documents are represented as nodes and citation relationships as edges. For our experiments, we used the largest connected component, which has a size of $|N| = 2120$.

**Synthetic Temporal Networks.** To extend our experiments, we used synthetically with community graphs with 2-block community networks with size of $60 \leq |N| \leq 120$ and 3-block community networks with size of $60 \leq |N| \leq 120$ .

## 4.2 BASELINE MODELS

We compared GraphK against both classical graph generation models and recent deep learning-based approaches. As traditional baselines, we included the **Erdős–Rényi (E-R)** model Erdös & Rényi (1959), which generates random graphs by connecting nodes with a fixed probability, and the **Barabási–Albert (B-A)** model Barabási et al. (2002), which produces scale-free networks using preferential attachment. Among deep learning-based methods, we evaluated **Graph-VAE** Simonovsky & Komodakis (2018), a variational autoencoder tailored for graph generation; **GraphRNN** You et al. (2018), which sequentially constructs graphs using a recurrent neural network to preserve topological structure, and **NetGAN**, a walk-based auto-regressive graph generation method. For diffusion-based models, we include **DiGress** Vignac et al. (2022) and **Pard** Zhao et al. (2024). Finally, as a scalable large graph generation method, we include **BiGG** Dai et al. (2020). This diverse set of baselines enables a comprehensive evaluation of our approach across both conventional and modern graph generation methods.

Table 2: Comparison of graph generation quality across models on synthetic (2-block Community) and real-world (Protein, CiteSeer) datasets. Metrics reported are MMD scores for spectral, orbit, and motif statistics (lower is better).

| | 2-Block Community | | | Protein | | | CiteSeer | | |
|---|---|---|---|---|---|---|---|---|---|
| | Spectral | Orbit | Motif | Spectral | Orbit | Motif | Spectral | Orbit | Motif |
| ER | 0.051 | 1.117 | 1.131 | 0.102 | 1.684 | 1.450 | 0.056 | 1.993 | 1.951 |
| BA | 0.044 | 1.214 | 1.420 | 0.100 | 1.139 | 1.284 | 0.048 | 1.127 | 1.383 |
| GraphVAE | 0.106 | 1.416 | 1.402 | 0.082 | 1.266 | 1.266 | 0.245 | 1.416 | 1.422 |
| GraphRNN | 0.061 | 1.151 | 1.116 | 0.301 | 0.752 | 0.804 | | OOM | |
| NetGAN | **0.021** | 0.298 | 0.398 | 0.075 | 1.112 | 1.172 | **0.024** | 1.351 | 1.184 |
| BIGG | 0.122 | 0.331 | 0.428 | 0.080 | 0.257 | 0.434 | 0.174 | 1.333 | 1.342 |
| DiGress | 0.022 | 0.268 | 0.244 | **0.032** | 0.262 | 0.482 | | OOM | |
| Pard | 0.070 | 0.250 | 0.349 | 0.033 | **0.253** | **0.201** | | OOM | |
| GraphK | 0.068 | **0.250** | **0.233** | **0.032** | 0.354 | 0.528 | 0.096 | **0.2511** | **0.5379** |

## 4.3 EXPERIMENTAL RESULTS

For each dataset, we compute Maximum Mean Discrepancy (MMD) Gretton et al. (2012) scores between generated and real graphs using spectral distances, orbit counts, and motif distributions. As shown in Table 2, our model, GraphK, consistently outperforms or remains competitive with baseline models across multiple datasets and metrics. On the 2-block community dataset, GraphK

achieves lower orbit and motif scores compared to other baselines, whereas NetGAN attains the lowest spectral score. In the protein dataset, Pard performs well on orbit and motif metrics, with GraphK remaining fairly competitive. For the CiteSeer dataset, GraphK achieves lower overall scores than BiGG and NetGAN, while GraphRNN, DiGress, and Pard fail to scale due to Out-Of-Memory (OOM) errors during training, highlighting the scalability limitations of some autoregressive methods. Overall, these results indicate that GraphK captures both global and local structural properties more effectively and performs better or comparably to both traditional and neural baseline models.

Here, we provide a visual comparison of graphs generated by our method GraphK against GraphRNN and GraphVAE, as shown in Fig. 2.

| | **Original** | **GraphK** | **GraphRNN** | **GraphVAE** |
|---|---|---|---|---|
| **2-B COM** | $|V|=80, |E|=382$ | $|V|=110, |E|=717$ | $|V|=90, |E|=470$ | $|V|=60, |E|=462$ |
| **3-B COM** | $|V|=100, |E|=606$ | $|V|=120, |E|=761$ | $|V|=74, |E|=267$ | $|V|=100, |E|=677$ |
| **Protein** | $|V|=327, |E|=899$ | $|V|=267, |E|=895$ | $|V|=288, |E|=747$ | $|V|=67, |E|=172$ |

Figure 2: Visual comparison of graphs generated by different models on synthetic community and real-world protein datasets.

We also provide a visual comparison of graphs generated by our method GraphK against GraphRNN and GraphVAE, as shown in Fig. 2. While results on community graphs are comparable across methods, GraphK better captures the structure of protein graphs. This improvement is due to GraphK's permutation invariance, which allows it to model protein graphs more effectively regardless of node ordering. In contrast, GraphRNN relies on a specific node order, which limits its performance on such datasets. Further evaluation with GRAN can be found in Appendix A.10.

**Computational Time Efficiency.** The encoder (Node2Vec or VGAE) runs in time roughly proportional to $O(E)$ (Node2Vec uses random walks over edges) or $\mathcal{O}(N + E)$ for a GNN encoder. The GMM fitting on the node embeddings is at most $\mathcal{O}(N \cdot (\text{mixture components}) \cdot \text{EM iterations})$, which for reasonably chosen component counts is manageable. Generation involves sampling $N'$ node embeddings (i.e., trivial cost) and building a KD-tree ($\mathcal{O}(N \log N)$) for neighbor search; connecting top-$k$ neighbors for each node is $\mathcal{O}(Nk \log N)$ with the tree. Typically $k$ is small (like tens), so this is effectively $\mathcal{O}(N \log N)$. Overall, the proposed method scales near-linearly in the number of nodes for generation, much better than quadratically. It can, in principle, generate graphs

much larger than those seen in training by sampling more points (i.e., upscaling), as presented in the appendix. By contrast, many prior models either struggle with large $N$ (GraphVAE, GraphRNN, GRAN) or require heavy computation (diffusion models). This is a major scalability advantage over methods. We leave the full details of quantitative results in the appendix. In terms of memory, storing the KD-tree is $O(N)$ and handling adjacency also $\mathcal{O}(N^2)$ in the worst case if fully connected, but since only $k$ neighbors per node are kept, the resulting graph has about $kN$ edges (linear in $N$). As a consequence, there is a performance trade-off: speed vs. granularity, as discussed in the limitations.

## 4.4 LARGE GRAPH GENERATION

We extend our experiments to graphs with larger sizes. Although there is no strict boundary, we describe a graph as large if it has more than $10,000$ nodes. As an ablation study, we train GraphK on a graph with exactly $10,000$ nodes and calculate the time for generating new graphs with various sizes between $5,000$ and $50,000$.

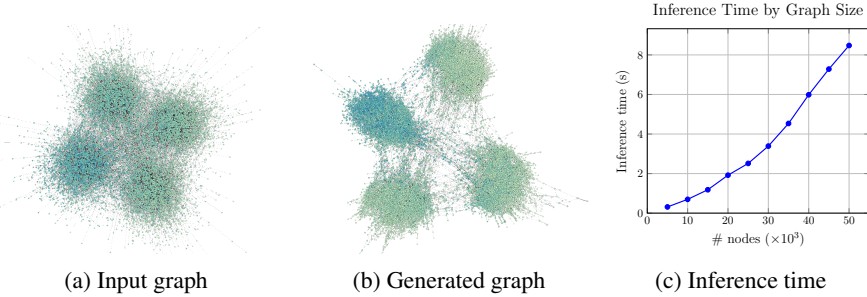

(a) Input graph      (b) Generated graph      (c) Inference time

Figure 3: Input graph shown in (a) has $10,000$ nodes and approximately $21,000$ edges. One of the generated graphs (b) has $15,000$ nodes and approximately $32,000$ edges. The parameter $k$ is selected as $12$. Subfigure (c) shows the decoding (inference) time in seconds with respect to the graph size.

For this experiment, Node2Vec is used as the encoder and trained for $800$ epochs, which took approximately 6 minutes. Number of GMM components is selected as 16, and $k$ as 12. GMM fitting took $1.46$ seconds, while sampling new embeddings from the learned distribution takes a negligible time (i.e., less than a second). Finally, decoding (i.e., inference) time with dot product is shared in Fig. 3c. The study shows that, technically, the proposed KDTree-based graph decoding approach allows the construction of large graphs having size up to $50,000$ nodes in less than 10 seconds.

Among the baseline models, GraphVAE, GraphRNN, DiGress, and Pard even struggle to process graphs with a thousand nodes. The inference time for BiGG, the only comparable baseline for large graph generation, is reported in their paper as approximately 7 minutes for $10,000$ nodes and about 20 minutes for $50,000$ nodes. Thus, for the dot-product decoder, the inference time of GraphK for $50,000$ nodes is 120 times faster than BiGG. However, it should be noted that, usage of the dot-product decoder requires a well-trained encoder with powerful representation capability. Otherwise, a learnable decoder (e.g., a neural network) can be utilized to increase the quality of generation, at the price of an increase in inference time by a constant factor (i.e., based on the cost of a single forward pass).

## 4.5 ASSESSING THE ENCODER FLEXIBILITY

In this ablation study, we aim to assess the generalizability of GraphK by using alternative graph encoders. Specifically, we utilize the Higher Order Proximity Preserved Embedding (HOPE) Ou et al. (2016) method, which focuses on directed graphs and seeks to learn two embedding vectors per node: one acting as a source and the other as a destination (see Fig. 4b).

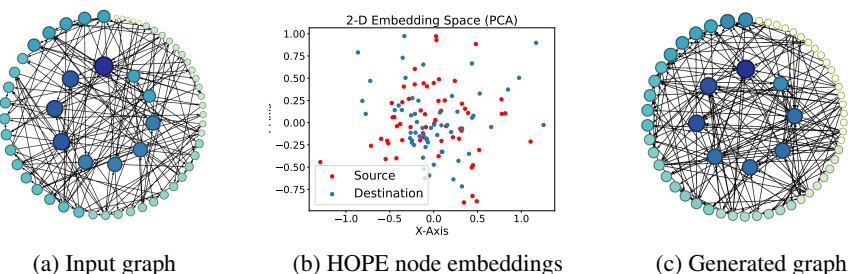

(a) Input graph      (b) HOPE node embeddings      (c) Generated graph

Figure 4: Directed graph generation with HOPE encoder. Darker nodes have higher in-degree.

For each node $v \in \mathcal{V}$, HOPE assigns latent embeddings $Z = \{z_v^{src}, z_v^{dst} \mid v \in \mathcal{V}\}$ that capture source and destination representations. We use separate GMMs to model these two vector sets and sample new embeddings. To adapt top-$k$ nearest neighbors retrieval into directed graph generation, we modify Eq. 6 as:

$$\mathsf{N}_k^{out}(v) = \operatorname{argmin}_{u \in \mathcal{V} \setminus \{v\}}^{(k)} \left\| z_v^{src} - z_u^{dst} \right\|_2 \tag{7}$$

Here, $\mathsf{N}_k^{out}(v)$ defines the outgoing neighbors (i.e., successors) of the node $v$. Similarly, the incoming neighbors (i.e., predecessors) $\mathsf{N}_k^{in}(v)$ can be obtained by swapping the roles of node $u$ and node $v$ within the distance calculation in Eq. 7. This formulation implies that there is a directed edge between node $v$ and node $u$ if the source vector of node $v$ resides close to the destination vector of node $u$.

For this ablation study, we used a directed E-R graph (Fig. 4a) as input. The generated output (Fig. 4c) is also a directed graph. For visualization, the dual circle layout has been chosen to help observe the distribution of node in-degree values, such that nodes with higher in-degree values are centered. The results confirm that GraphK is highly adaptable to diverse graph types when an appropriate encoder is utilized, and emphasize the flexibility of the proposed framework.

### LIMITATIONS

A key limitation of our approach lies in the smoothness assumption made during the decoder step. This assumption suggests that nodes with similar features are likely to be connected, which can limit ability of the model to capture all relevant relationships in the graph. Specifically, two nodes that may not appear in the top-$k$ rankings based on feature similarity might still have a very important edge between them, which our model could overlook. This limitation highlights the need for further research into relaxing the smoothness assumption or incorporating more sophisticated mechanisms that can better identify crucial connections beyond simple feature-based similarity. Another limitation is that the embedding preserves local structure, i.e, nodes that were connected or in the same community in the original graph will lie close together in latent space, so reconnecting by proximity yields a similar topology. This strategy avoids evaluating all node pairs for edge probabilities; instead of a complete $O(N^2)$ pairwise comparison, a KD-tree helps find nearest neighbors efficiently. The trade-off is that each node ends up with approximately k edges, i.e., the resulting degree distribution and adjacency are thus constrained by the geometric layout. This can naturally capture clustering (latent clusters lead to dense connections inside a community) but might struggle to generate graphs with grid structure or line.

## 5 CONCLUSION

We proposed **GraphK**, a graph generation method that effectively captures both structural properties and permutation-invariant characteristics of graphs. Through extensive experiments on synthetic and real-world datasets, including community graphs, protein structures, and citation networks, we demonstrated that GraphK achieves superior performance in terms of spectral, orbit and motif metrics compared to both traditional and deep learning-based baselines. GraphK not only preserves key topological features such as spectral distributions, orbits, and motifs, but also maintains robustness to node ordering.

## REPRODUCIBILITY STATEMENT

The proposed GraphK is fully reproducible. The model components and training pipeline are specified in Section 3 and Fig. 1 (encoder/sampler/decoder), with an explicit note that all data and code will be released; our camera-ready will include an anonymous repository with scripts to reproduce every table and figure. For implementation details that affect outcomes, Appendix A.2 gives the Node2Vec objective and softmax, Appendix A.3 provides the complete EM updates for the GMM sampler, and Appendix A.4 lists the concrete hyperparameters and optimizer settings we used. Our datasets and preprocessing choices are summarized in Section 4.1 and Appendix A.4.1, and the baselines we compare against are enumerated in Section 4.2. We report evaluation protocols and metrics (MMD over spectra, orbits, and motifs) in Section 4.3 and Table 2, enabling like-for-like replication. We also document analyses that probe specific claims: permutation-invariance stress tests in Appendix A.5, upscaling/downscaling behavior in Appendix A.6. Finally, Appendix A.9 details our power-law study on Gnutella05, including degree histograms for original vs. generated graphs.

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

# A  APPENDIX

## A.1  NOTATIONS

Table 3 summarizes the notations used throughout the paper.

Table 3: Summary of Notations

| Symbol | Description |
|---|---|
| $\mathcal{G} = (\mathcal{V}, \mathcal{E}, \mathcal{X})$ | Input graph with node set $\mathcal{V}$ and edge set $\mathcal{E}$ and features of nodes $\mathcal{X}$ |
| $n = |\mathcal{V}|$ | Number of nodes in the graph |
| $d$ | Dimensionality of the latent space |
| $z_v \in \mathbb{R}^d$ | Latent representation of node $v \in \mathcal{V}$ |
| $Z = \{z_v \mid v \in \mathcal{V}\}$ | Set of latent embeddings for all nodes |
| $\|z_v - z_u\|_2$ | Euclidean distance between node embeddings $z_v$ and $z_u$ |
| $\mathcal{N}_k(v)$ | Top-$k$ nearest neighbors of node $v$ in the latent space |
| $k$ | Number of neighbors retrieved (set to mode of degree distribution) |
| $\mathrm{argmin}^{(k)} \|z_v - z_u\|_2$ | Operator that returns the $k$ nodes closest to $z_v$ |
| $f_\theta(z_v, z_u)$ | Learnable decoder function predicting edge strength or existence |

## A.2  NODE2VEC ENCODER DETAILS

Node2Vec Grover & Leskovec (2016) learns node embeddings by simulating biased random walks on the graph and applying the Skip-Gram model. For each node $v \in V$, Node2Vec optimizes the following objective:

$$\max_\Phi \sum_{v \in V} \sum_{u \in \mathbf{N}(v)} \log \Pr(u \mid \Phi(v)), \tag{8}$$

where $\Phi : V \rightarrow \mathbb{R}^d$ is the embedding function, $\mathbf{N}(v)$ denotes the neighborhood of node $v$ obtained through biased random walks, and $\Pr(u \mid \Phi(v))$ is modeled via a softmax:

$$\Pr(u \mid \Phi(v)) = \frac{\exp(\Phi(u)^\top \Phi(v))}{\sum_{v' \in V} \exp(\Phi(v')^\top \Phi(v))}. \tag{9}$$

Here, $\Pr(u \mid \Phi(v))$ denotes the probability of node $u$ appearing in the context of node $v$, given their embeddings. It reflects how likely $u$ is to co-occur with $v$ based on the similarity of their vector representations.

## A.3  EXPECTATION–MAXIMIZATION FOR GMM

The EM algorithm proceeds iteratively with the following two steps:

**E-step.** We compute the posterior responsibility $\gamma_{ik}$, which represents the probability that the data point $z_i$ was generated by the $k$-th Gaussian component:

$$\gamma_{ik} = \frac{\pi_k \mathcal{N}(z_i \mid \mu_k, \Sigma_k)}{\sum_{j=1}^K \pi_j \mathcal{N}(z_i \mid \mu_j, \Sigma_j)}. \tag{10}$$

**M-step.** We update the parameters of the mixture model based on the computed responsibilities:

$$\pi_k^{\text{new}} = \frac{1}{N} \sum_{i=1}^N \gamma_{ik}, \tag{11}$$

$$\mu_k^{\text{new}} = \frac{\sum_{i=1}^N \gamma_{ik} z_i}{\sum_{i=1}^N \gamma_{ik}}, \tag{12}$$

$$\Sigma_k^{\text{new}} = \frac{\sum_{i=1}^{N} \gamma_{ik}(z_i - \mu_k)(z_i - \mu_k)^{\top}}{\sum_{i=1}^{N} \gamma_{ik}}. \tag{13}$$

These E and M steps are repeated until convergence. Once the model is trained, we sample a component $k \sim \text{Categorical}(\pi_1, ..., \pi_K)$ and then draw node representations $\tilde{z} \sim \mathcal{N}(\mu_k, \Sigma_k)$, enabling diverse and structured node embeddings for graph generation.

## A.4 IMPLEMENTATION DETAILS

### A.4.1 DATA PREPROCESSING

Before training and evaluation, we apply several standard preprocessing steps to ensure consistency across datasets. First, graphs are loaded from pre-specified pickle files and renumbered to contiguous node indices, then converted to PyTorch Geometric `Data` objects with a constant node feature vector of dimension one. For supervised training of the edge decoder, positive edges are paired with some ratio of negatives drawn via negative sampling, and binary edge labels are constructed. During generation, we threshold the predicted adjacency, remove self-loops, drop isolated nodes, and optionally retain only the largest connected component to ensure that outputs are well-formed and comparable. For evaluation, we normalize distributions of Laplacian spectra before computing MMD scores, and we exclude empty graphs from consideration. Motif and orbit statistics are obtained through consistent node reindexing and external ORCA calls. Together, these steps standardize input graphs, ensure realistic generated outputs, and enable fair and reproducible evaluation.

### A.4.2 EXPERIMENTAL SETUP

We evaluated deep graph generation baselines using their official implementations and default settings: GraphRNN, NetGAN, and DiGress. Since NetGAN relies on an older version of TensorFlow (1.x), we ran it locally, while the other models were executed in Google Colab notebooks for convenience. For evaluation, each method generated 20 graphs, and we measured their similarity to the original graph using the MMD distance. The same evaluation procedure was applied consistently across all baselines.

### A.4.3 IMPLEMENTATION DETAILS OF MODEL COMPONENTS

**Encoder.** For the Node2Vec encoder, the choice of hyperparameters depends on the input graph characteristics and size. However, we commonly used a walk length of 20, a context size of 10, and a number of walks to sample per node of 10. Additionally, we used an embedding size of 16. We used second-degree bias parameters p and q as 0.5 and 2.0, respectively. This choice is to exploit a higher degree of neighborhood. Especially for the community graphs, we observed that having a light encoder training is useful since a tight one causes an embedding space with tightly clustered nodes, resulting in missing the inter-community edges. We set the learning rate as 0.001 learning rate trained a Node2Vec model for 600 epochs. In addition to Node2Vec, we conducted experiments with a VGAE as the encoder. In these experiments, we used a number of input channels as 16, hidden channels as 32, and latent size as 16. As optimizer, we used AdamW and set the learning rate as 0.001. Because of its flexibility, ease of usage, and ability to capture localities well, we set our default encoder as Node2Vec.

**Sampler.** GMM sampler requires a critical hyperparameter, number of components, which refers to the number of Gaussian distributions that the model will use to represent the data (i.e., node embeddings). This hyperparameter should be chosen carefully so that it does not oversimplify or undersimplify the underlying graph distribution in Euclidean space. For example, in the case of graphs with clear communities and rare inter-community edges, we observed that setting the number of components hyperparameter equal to the number of communities is useful. However, this manual approach is not feasible when the underlying graph distribution is complex or communities are unclear (e.g., protein graphs). In those cases, we used the Variational Bayesian Gaussian Mixture Model (VBGMM), a Bayesian extension of traditional GMM. Instead of pre-specifying the number of Gaussian distributions, VBGMM can automatically infer it, resulting in regularized and softer assignments of data points to clusters.

**Decoder.** For the decoding stage, we used a simple binary classifier composed of a neural network with two linear layers. Since the node embeddings are already initialized using the encoder's output, using a more complex decoder could lead to overfitting. To convert pairs of node embeddings into edge features, we concatenated several metrics: the Hadamard product, L1 distance, L2 distance, and the element-wise average.

We trained the decoder using the AdamW optimizer with a learning rate of $0.001$ and a small weight decay. Instead of freezing the encoder-generated node embeddings, we fine-tuned them during training with a much smaller learning rate (e.g., $10^{-5}$). Since link prediction is usually a local task, this fine-tuning can be particularly beneficial when the encoder focuses more on global structural properties of graph (such as GraphWave), which lack locality.

### A.5 PERMUTATION INVARIANCE EVALUATION

We conducted an additional experiment to quantify the effect of node ordering on graph generation model outputs. The simplest approach for evaluating permutation invariance involves taking a single input graph $G$ and producing $P$ variants by randomly shuffling the node indices, effectively applying different permutations $\pi$ to its adjacency matrix. While topics such as graph isomorphism and canonical graph representations form a separate research area in graph theory Huang et al. (2022); Ma et al. (2024), our focus here is on extending this analysis to real datasets whose inherent structural properties make them particularly sensitive to node ordering.

With this goal in mind, grid graphs initially appear ideal due to their high symmetry, regular topology, large number of structurally indistinct nodes, and extensive automorphism groups. However, grid graphs are outside the modeling scope of our current framework (as discussed in the Limitations section). On the other hand, stochastic block community graphs are overly forgiving. Once a traversal begins within a dense community, breadth-first or depth-first searches tend to collapse multiple permutations into very similar, block-oriented sequences, i.e., quickly traversing one clique before moving to another, which conceals potential ordering biases. We therefore selected Protein Dobson & Doig (2003) graphs for this analysis. Their polymer-backbone topology, characterized by long self-avoiding paths, short secondary-structure cycles, and recurrent local motifs, provides sufficient symmetry to highlight ordering bias clearly.

Accordingly, in this part of our evaluation, we compare our model exclusively against GRAN Liao et al. (2020), the current state-of-the-art baseline known for its consistently strong performance across various graph generation benchmarks. We intentionally exclude diffusion-based models from the comparison, as they are not well-suited for large graphs and exhibit impractical runtimes in benchmark studies, placing them outside the scope of this competitive evaluation Zhao et al. (2024). To assess how the impact of permutation invariance scales with graph size, we divided our dataset into four groups based on node counts: Group 1 (100–200 nodes), Group 2 (200–300 nodes), Group 3 (300–400 nodes), and Group 4 (400–500 nodes). For each generated graph, we computed the Laplacian-spectrum MMD metric. This choice was deliberate, as the Laplacian spectrum (i.e., the set of eigenvalues of the Laplacian matrix) remains invariant to node permutations. Additionally, the Laplacian spectrum effectively captures global structural properties, unlike purely local descriptors such as node degrees, orbit distributions, or clustering coefficients O'Bray et al. (2021). Although these local metrics are also valuable in assessing graph quality, as discussed previously in Table 2, they may remain relatively unchanged despite significant overall topological alterations. A visual comparison was provided in Fig. 10a, here presented with additional detailed results in Fig. 5.

In smaller graphs (under 300 nodes), GRAN achieves slightly superior performance, reproducing spectral characteristics marginally better than GraphK (approximately 15% lower). At this scale, GRAN's block attention mechanism effectively explores multiple node permutations without accumulating noticeable biases. However, the critical crossover point at Group 3 supports our hypothesis. As graph size increases beyond 300 nodes, GRAN's Spectral MMD significantly deteriorates, rising from $1.9 \times 10^{-2}$ (300–400 nodes) to $2.5 \times 10^{-2}$ (400–500 nodes). This sharp increase highlights the growing negative impact of node-order sensitivity as graphs become larger and structurally more complex. In contrast, GraphK demonstrates strong permutation invariance, achieving a 52% lower Spectral MMD compared to GRAN at the largest graph size (400–500 nodes).

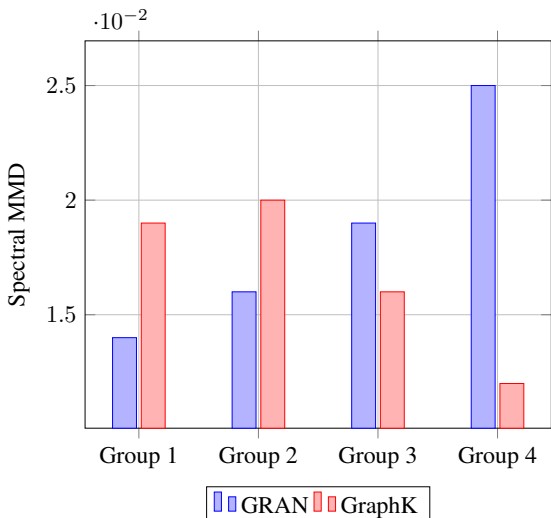

Figure 5: GraphK vs GRAN in protein dataset. Group 1: (100-200 nodes), Group 2: (200-300 nodes), Group 3: (300-400 nodes), Group 4: (400-500 nodes)

### A.6 UPSCALING - DOWNSCALING PERFORMANCE

To evaluate the upscaling and downscaling capabilities of GraphK, we conducted an experiment using a three-block community graph. We intentionally set one community to be larger than the others, with approximate sizes of 200, 100, and 100 nodes. Using our proposed GraphK model, we upscaled the graph by a factor of 10, as shown in Fig. 6b. Visual inspection reveals that one community remains slightly larger than the others, demonstrating GraphK's ability to preserve the relative sizes of communities during upscaling. Next, we trained GraphK on the upscaled graph and generated graphs having a similar number of nodes as the original graph (i.e., graph shown in Fig. 6a). The results, depicted in Fig. 6c, highlight GraphK's effectiveness in downscaling while maintaining structural properties.

The ability to upscale graphs is particularly critical when available real-world data is sparse, limited, or sensitive. These situations are commonly encountered in domains like social network privacy analysis Abawajy et al. (2016), network traffic simulation Li et al. (2024), and resilience testing of large-scale infrastructure systems. Specifically, in applied machine learning studies, GraphK's upscaling capability might enable effective data augmentation, providing richer synthetic datasets for training robust graph-based machine learning models Zhou et al. (2025). Furthermore, it can facilitate scenario testing and forecasting in network science, where examining structural evolution at scale is crucial yet difficult with original, fixed-size datasets.

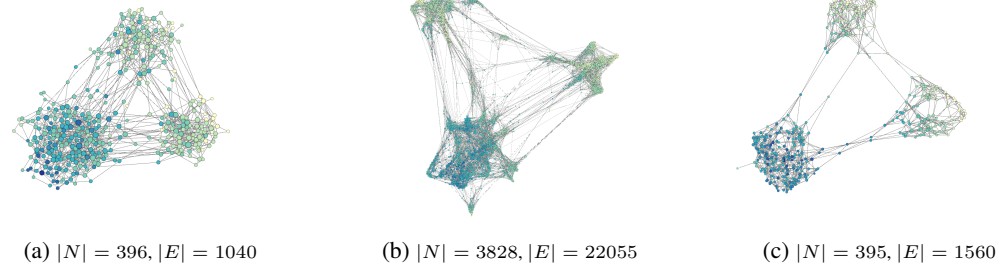

(a) $|N| = 396, |E| = 1040$      (b) $|N| = 3828, |E| = 22055$      (c) $|N| = 395, |E| = 1560$

Figure 6: Figure (a) shows a graph with three communities generated using the Stochastic Block Model. After upscaling, we obtain Figure (b). Then, we downscale it back, resulting in Figure (c).

### A.7 VISUAL ASSESSMENT

We provide additional visual examples by GraphK, emphasizing its ability to generate graphs with varying sizes but similar structural properties.

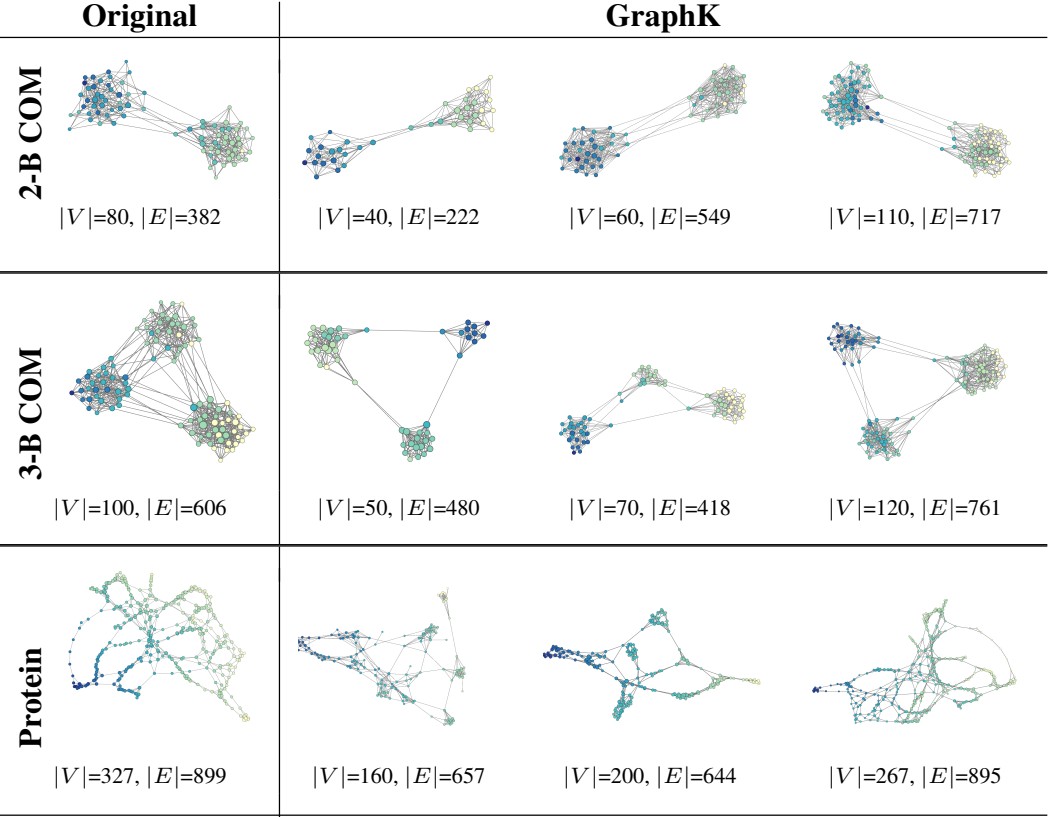

Figure 7

### A.8 NODE CLASSIFICATION DOWNSTREAM TASK

To evaluate the effectiveness of the embeddings sampled from GMM, we conducted an experiment combining the encoder and sampler for node embedding augmentation in a node classification task on the CiteSeer dataset. First, we obtained Node2Vec embeddings for all 3,327 nodes. These embeddings were then split into training (120 nodes), validation (500 nodes), and test (1,000 nodes) sets, and an XGBoost Chen & Guestrin (2016) classifier was trained on the training set. Next, we fitted a Gaussian Mixture Model (GMM) with 6 components to the training embeddings and generated additional node embeddings by sampling from the GMM. The sampled embeddings were assigned labels using a K-Nearest Neighbors Cover & Hart (1967) classifier. Finally, we retrained the XGBoost model with the augmented training set and evaluated performance improvements. The results, shown in Fig. 8, indicate that augmenting the training embeddings via GMM sampling can increase classification accuracy by 2% to 10%.

### A.9 GENERATING POWER-LAW BEHAVIOR

At first glance, the fixed neighborhood size $k$ might appear to limit the emergence of a heavy-tailed degree distribution. However, this is not necessarily the case. To build intuition, consider a point $P$ near the center of an embedding space containing 99 other points, with $k = 5$. While $P$ selects only 5 neighbors, it may be chosen as a neighbor by many other points, such as 50. In an undirected graph, this would result in $deg(P) = 55$. In other words, although each node can select at most $k$

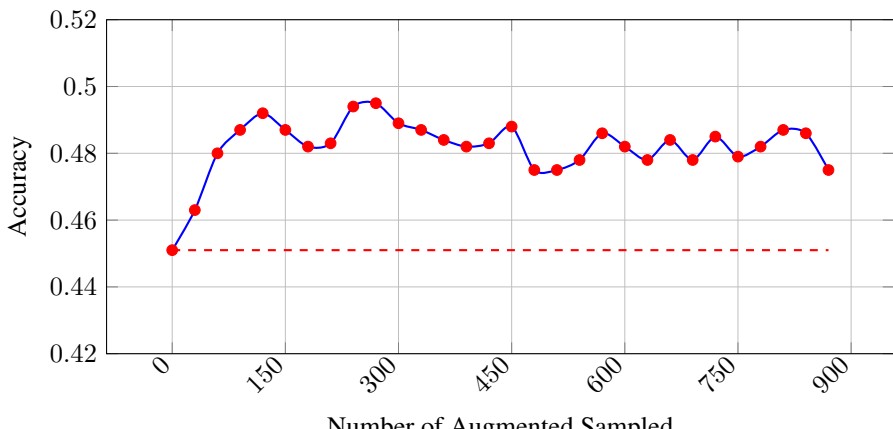

Figure 8: Augmentation effect on node classification task (Dashed line is the accuracy without augmentation.)

neighbors, it may be selected by many nodes. This asymmetric selection may lead to the possibility of high-degree hubs and, consequently, a power-law-like degree distribution.

To illustrate this phenomenon, we experimented with the Gnutella05 ($|V| = 8846, |E| = 31839$) peer-to-peer file-sharing network, which exhibits a power-law-like degree distribution. During this experimentation, we set k=18 and obtained the highest degree of 54. As seen in the Fig. 9, the GraphK-generated graph ($|V| = 8578, |E| = 32667$) has a clear power-law-like distribution, where the majority of nodes have a low degree while a few nodes act as high-degree hubs. This experiment shows that GraphK can generate graphs exhibiting a power-law degree distribution, as commonly observed in real-world networks.

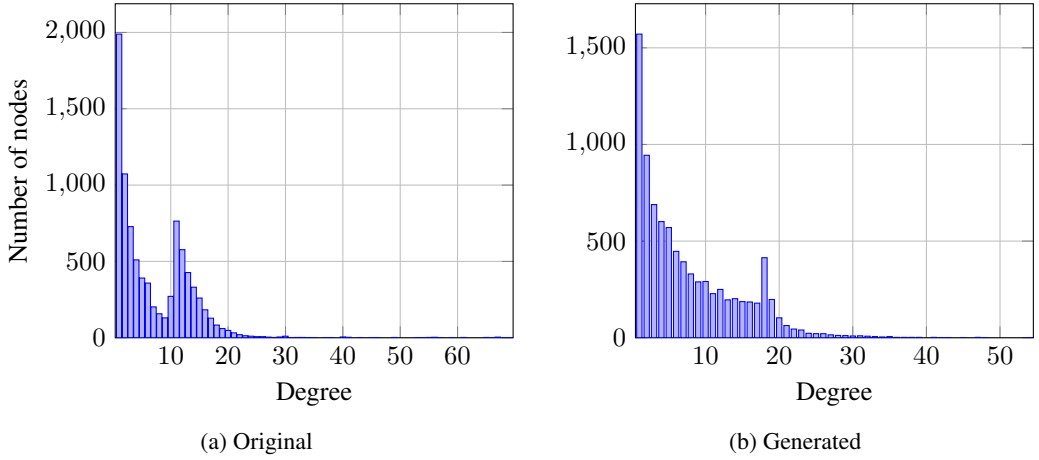

(a) Original          (b) Generated

Figure 9: Comparison of two histograms

## A.10    ADDITIONAL PERMUTATION INVARIANCE ASSESSMENT

We also evaluated GRAN using the spectral metric on the Protein dataset. As shown in Fig. 10a, GRAN, which is not permutation invariant, achieves a spectral score of 0.038, while our method, GraphK, achieves a lower score of 0.032. This difference highlights the advantage of permutation invariance in our method, particularly because the spectral metric is sensitive to node orderings. Models like GRAN that rely on fixed node permutations tend to perform worse on datasets where node identities are not aligned. Additionally, we examined the impact of varying the parameter $k$ in the KDTree algorithm as shown in as shown in Fig. 10b. As the number of generated nodes

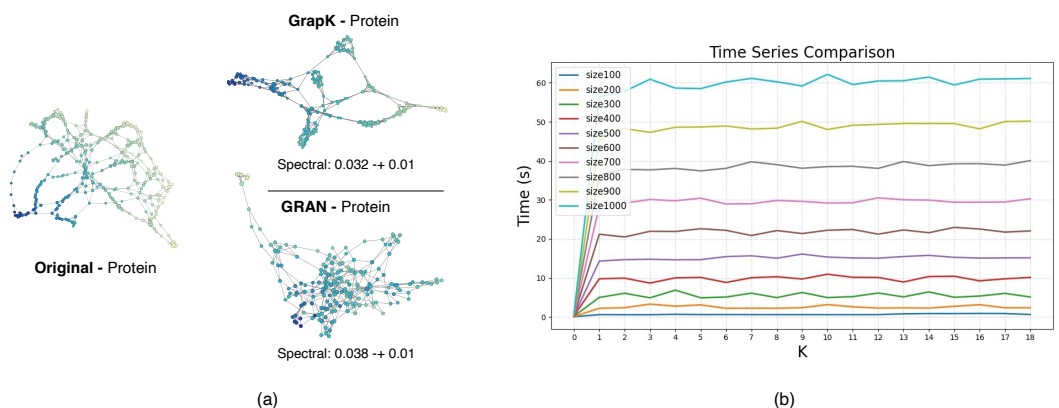

(a)                                                                      (b)

Figure 10: Comparison of spectral scores on the Protein dataset, highlighting the impact of permutation invariance (a), and decoder time calculation to show scale (b)

increases, the decoder time grows; however, after a certain point, the runtime stabilizes due to the $\mathcal{O}(n \log n)$ complexity of the KDTree.

