# OpenReview forum: "Beyond Autoregression: Permutation-Invariant Graph Generation with Scalable Edge Construction"
_ICLR.cc/2026/Conference — ICLR 2026 Conference Desk Rejected Submission_

### Official Review · Reviewer_FTyv · 2025-10-20

**Soundness:** 2
**Presentation:** 3
**Contribution:** 2
**Rating:** 2
**Confidence:** 4

**Summary:**

This paper proposes GraphK, a novel permutation-invariant and scalable graph generation framework.


GraphK follows a modular encoder–sampler–decoder architecture:

Encoder: Maps the input graph into a latent space using either Node2Vec or VGAE embeddings.
Sampler: Fits a Gaussian Mixture Model (GMM) on these node embeddings to enable scalable sampling, allowing upscaling or downscaling of graph size.
Decoder: Generates edges using a k-nearest neighbor (KDTree) search in the latent space.

GraphK claims three major advantages: 1) Permutation invariance (independent of node ordering)
2) Scalability (generation beyond training graph sizes), and
3) Computational efficiency (near-linear complexity).

Experiments on synthetic community graphs, protein datasets, and CiteSeer show that GraphK achieves competitive or superior performance to GraphVAE, GraphRNN, and NetGAN under MMD-based structural similarity metrics (spectral, orbit, motif).

While the paper is well written, clearly structured, and easy to follow, the experimental and theoretical evidence supporting its main claims remains limited.

* Limited Comparison to Scalable Deep Baselines:
While the paper compares against GraphRNN, NetGAN, and DiGress, it omits scalable graph generators such as **BiGG [1]**, which achieves state-of-the-art structure preservation with memory cost 𝑂(ElogN). Including BiGG baselines would enable a comprehensive evaluation.

* Permutation Invariance:
Although GraphK emphasizes permutation-invariant graph generation, the paper provides no formal proof or justification of this property. In particular, it is unclear how the gradient updates with respect to parameters remain invariant under node permutations, especially given that the Node2Vec encoder is not strictly permutation invariant (its random-walk–based embeddings depend on node identities).

* Scalability:
The framework is theoretically claimed to scale to graphs with >10k of nodes [table 1] due to its 𝑂(𝑁log𝑁) decoding complexity. However, the largest empirical test involves graphs of only about 4,000 nodes. Demonstrating results on large graphs (>10K nodes) would provide stronger empirical support for this scalability claim.


* No Ablation on Encoder Type:
The paper claims flexibility in encoder choice (Node2Vec, VGAE, etc.), yet no ablation or quantitative analysis is presented to validate this claim. Since the encoder choice critically influences the latent representation and downstream sampling, an encoder ablation would strengthen the paper claim.

* Limited Dataset:
The experiments are limited to small and medium-sized graphs (protein, community, and CiteSeer datasets). Expanding to widely used benchmarks such as grid and lobster graphs—used in prior works including BiGG and GraphRNN would better demonstrate the generality of the proposed framework.

[1] Dai, Hanjun, et al. "Scalable deep generative modeling for sparse graphs." International conference on machine learning. PMLR, 2020.

**Strengths:**

Clear and Modular Framework

Novel Latent Sampling Strategy

**Weaknesses:**

No End-to-End Training Objective

Limited empirical and theoretical support

Smoothness Assumption in Decoder

**Questions:**

How is the encoder fine-tuned during training. Is it frozen after pretraining, or updated jointly with the edge decoder?

How sensitive is GraphK’s performance to the k parameter in KDTree edge construction?

---

> ### Author Response · Authors · 2025-11-20
>
> 1- “Limited Comparison to Scalable Deep Baselines: While the paper compares against GraphRNN, NetGAN, and DiGress, it omits scalable graph generators such as BiGG [1], which achieves state-of-the-art structure preservation with memory cost 𝑂(ElogN). Including BiGG baselines would enable a comprehensive evaluation.”
> Thank you very much for your suggestion. We have added BiGG as an additional baseline and results can be seen in the rebuttal version.
>
> 2- “Permutation Invariance: Although GraphK emphasizes permutation-invariant graph generation, the paper provides no formal proof or justification of this property. In particular, it is unclear how the gradient updates with respect to parameters remain invariant under node permutations, especially given that the Node2Vec encoder is not strictly permutation invariant (its random-walk–based embeddings depend on node identities).”
>
> The permutation invariance is achieved by the Gaussian Mixture Model (GMM) sampler, thanks to its ability to learn the global distribution of the set of node embeddings. It behaves like an aggregator that compresses all node embeddings into a single probability distribution describing the overall statistical profile of the graph. Thus, shuffling the order of nodes does not change the learned GMM parameters or generated graph. Since the permutation-invariance is recognized during GMM fitting, it does not matter whether the encoder is permutation-invariant or not. This is one of the reasons we emphasize the flexibility of the framework (i.e., any node embedding extraction algorithms including GNN-based or traditional can be utilized).
>
> 3- “Scalability: The framework is theoretically claimed to scale to graphs with >10k of nodes [table 1] due to its 𝑂(𝑁log𝑁) decoding complexity. However, the largest empirical test involves graphs of only about 4,000 nodes. Demonstrating results on large graphs (>10K nodes) would provide stronger empirical support for this scalability claim.”
>
> Thank you for pointing this out. We agree that evaluating the framework on larger graphs is important. In fact, the largest graph we used is the Gnutella peer-to-peer file-sharing network (shown in Appendix 10), which contains more than 8k nodes and 31k edges. Although this section primarily focuses on modeling the degree distribution, we will move this content from the appendix into the main manuscript to improve clarity and visibility. Additionally, we added a new large-scale graph (10k nodes and 20k edges) and generated new graphs with even larger sizes (15k nodes and 32k edges). The results are added to the rebuttal version along with train and inference times.
>
> 4- “No Ablation on Encoder Type: The paper claims flexibility in encoder choice (Node2Vec, VGAE, etc.), yet no ablation or quantitative analysis is presented to validate this claim. Since the encoder choice critically influences the latent representation and downstream sampling, an encoder ablation would strengthen the paper claim.”
> Thank you very much for this suggestion. Since we added two types of encoder, we did not include more. We have added a new encoder HOPE[1] in the rebuttal version.
>
> 5- “Limited Dataset: The experiments are limited to small and medium-sized graphs (protein, community, and CiteSeer datasets). Expanding to widely used benchmarks such as grid and lobster graphs—used in prior works including BiGG and GraphRNN would better demonstrate the generality of the proposed framework.”
>
> Thank you for pointing the importance of graph data variety out. Both intuitively and experimentally, we observed that the introduced framework is more suitable to graphs with irregular sub-structures (e.g., community), rather than regular patterns such as grid or line (see the Limitations section).
>
> [1] Mingdong Ou, Peng Cui, Jian Pei, Ziwei Zhang, and Wenwu Zhu. 2016. Asymmetric Transitivity Preserving Graph Embedding. In Proceedings of the 22nd ACM SIGKDD International Conference on Knowledge Discovery and Data Mining (KDD '16).

---

### Official Review · Reviewer_o1tc · 2025-11-03

**Soundness:** 3
**Presentation:** 3
**Contribution:** 2
**Rating:** 6
**Confidence:** 2

**Summary:**

This paper proposes a novel graph generation framework named GraphK, which aims to address three key challenges in current deep graph generation models: the lack of permutation invariance, the difficulty in generating graphs of sizes different from the training set, and high computational costs. GraphK adopts an "encoder-sampler-decoder" paradigm. It achieves permutation-invariant, size-flexible, and computationally efficient graph generation by mapping graphs into a latent space, sampling new node embeddings within this latent space, and efficiently reconstructing edges based on geometric proximity.

**Strengths:**

1. The integration of a permutation-invariant encoder, GMM-based latent space sampling, and efficient geometric decoding within a unified framework is conceptually clear and well-structured. This approach cleverly decouples graph structure learning from size generation, offering a novel solution for size-agnostic graph generation.

2. The paper provides a detailed time complexity analysis for each stage of GraphK, explicitly stating that the generation process has a near-linear complexity of O(N log N). This represents a significant potential advantage compared to many autoregressive or diffusion models with quadratic complexity.

**Weaknesses:**

1. The baseline comparisons in Table 2 are not sufficiently comprehensive. The most recent baseline included is from 2022. The authors should, at a minimum, compare against more recent methods listed in Table 1, such as EDP-GNN and GDSS.

2. While the authors claim the method is efficient and provide a theoretical time complexity analysis, there is a lack of comparison experiments with other methods regarding practical metrics like actual training time and inference overhead.

3. The paper claims the method is scalable, but the experiments primarily rely on synthetic data and medium-scale graphs. It remains unclear whether GMM fitting would become a computational bottleneck on larger-scale graphs.

**Questions:**

See Weaknesses.

---

> ### Author Response · Authors · 2025-11-20
>
> 1- “The baseline comparisons in Table 2 are not sufficiently comprehensive. The most recent baseline included is from 2022. The authors should, at a minimum, compare against more recent methods listed in Table 1, such as EDP-GNN and GDSS.”
>
> As we stated in the response of reviewer #7Utq, EDP-GNN and GDSS work on small graphs (up to 400 nodes) which is not applicable for our datasets (up to 10k nodes). But instead, we included two more baselines to our comparison (PARD[1], BiGG[2]). Our results show that GraphK outperforms (and competitive on some metrics) PARD and BiGG(can be seen in the rebuttal revision) .
>
> 2- “While the authors claim the method is efficient and provide a theoretical time complexity analysis, there is a lack of comparison experiments with other methods regarding practical metrics like actual training time and inference overhead.”
> Thank you very much for your comment. We actually have added training and inference time of GraphK in Figure 6 (line 874), though we optimised a lot to reduce the run time and inference time (can be seen in the rebuttal revision). We also added other methods run times.
>
> 3- “The paper claims the method is scalable, but the experiments primarily rely on synthetic data and medium-scale graphs. It remains unclear whether GMM fitting would become a computational bottleneck on larger-scale graphs.”
> We have added training and inference time from 1k to 30k including encoder embedding model, GMM fitting and KDTree Search. It can be seen in the rebuttal version. Other methods work on graphs up to node size of approximately 2k.
>
> [1] Zhao, Lingxiao, Xueying Ding, and Leman Akoglu. "Pard: Permutation-invariant autoregressive diffusion for graph generation." Advances in Neural Information Processing Systems 37 (2024): 7156-7184 [2] Dai, Hanjun, et al. "Scalable deep generative modeling for sparse graphs." International conference on machine learning. PMLR, 2020.

---

> > ### Comment · Reviewer_o1tc · 2025-11-26
> >
> > Thank you for the author’s response. I have decided to keep my original score.

---

### Official Review · Reviewer_Kr27 · 2025-11-03

**Soundness:** 2
**Presentation:** 3
**Contribution:** 2
**Rating:** 2
**Confidence:** 4

**Summary:**

This paper introduces a method in the graph generation field that has three main parts: encoder, sampler, and decoder. Their key contributions are:
- The generation process does not depend on how nodes are labeled or ordered in the input.
- By fitting a GMM to the node embeddings and flexible sampling, the model allows the generation of larger or smaller graphs while preserving key structural patterns from the original input.
- By constructing a decoder based on a k-nearest neighbor strategy and constructing a KD-tree on the sample latent embeddings.

**Strengths:**

- This paper contributes by confronting two issues in graph generation: scalability and ensuring permutation invariance. The authors dedicate significant effort to reducing the notorious computational complexity in this domain.
- The core strength lies in the novel, three-part GraphK framework. Encoder: The Encoder is remarkably flexible. By intelligently blending both shallow and deep learning encoders, the model is well-equipped to handle a variety of graph types and structural complexities. Sampler that uses a Gaussian Mixture Model (GMM) to model diverse node representations.
- The decoder component integrates a KD-tree for highly efficient nearest-neighbor search, aiming to address the computational complexity.

**Weaknesses:**

- The experiments conducted are severely limited (Table 2, Figures 2 and 3). The comparison methods are highly restricted and mostly do not reflect the Current State-of-the-Art, with the proposed approach failing to be benchmarked against any of the state-of-the-art methods from recent years (especially those from 2024 and 2023).
- In Figure 2, which aims to compare models in generating graphs similar to the original graph, no evaluation metric is introduced; the comparison relies solely on the visual appearance of the graphs. Furthermore, the proposed model is compared only against two other models in this section, which are Not Reflective of the Current State-of-the-Art.
- The claim regarding the flexibility of the Encoder in handling various graph types and structures was not adequately investigated in the experiments, and the results do not convincingly demonstrate this versatility.
- The claim of being scalable is not sufficiently covered in the experimental section. It is expected that the proposed method will be compared against State-of-the-Art methods explicitly designed for scalable graph generation.
- In the Decoder section, the manifold smoothness assumption is introduced but presented without any explanation or theoretical justification as to why this assumption is logical or valid for the proposed method. It was expected that theoretical reasons and proofs would be provided to support this critical assumption.

**Questions:**

According to the identified weaknesses, it is recommended that the following questions and suggestions be addressed:
- Address the limited number of experiments and the lack of use of current, state-of-the-art methods in Table 2 and Figures 2 and 3.
- Design experiments specifically to demonstrate the claim regarding the flexibility of the encoder.
- Introduce a quantitative metric for the comparisons presented in Figure 2, and do not rely solely on the visual images of the output graphs from each method.
- Design experiments for a more rigorous investigation of scalability, specifically through comparison with state-of-the-art methods.
- Could you explain more about the smooth manifold assumption, including why this assumption is reasonable in the context of the proposed method, and try to offer theoretical proof for it?

---

### Official Review · Reviewer_7Utq · 2025-11-04

**Soundness:** 3
**Presentation:** 2
**Contribution:** 2
**Rating:** 2
**Confidence:** 2

**Summary:**

The authors propose GraphK, which addresses the generalization issues in graph generation models through structural flexibility and computational efficiency. GraphK supports generating graphs with an arbitrary number of nodes. Experiments on multiple datasets demonstrate that GraphK outperforms existing methods.

**Strengths:**

Originality. The paper is original, and the capability to support upscaling is interesting.

Quality. The manuscript is of high quality, with rich and comprehensive experimental results.

Clarity. I fully understand what the authors are conveying—the writing is clear and well-structured.

Significance. I remain somewhat skeptical about the paper's overall significance. Beyond introducing a graph generation method that supports upscaling, I cannot understand additional substantial contributions. I will elaborate on this concern in the Question and Weaknesses sections.

**Weaknesses:**

1.Upsampling appears to be a popular approach in graph generation (please correct me if I'm mistaken). The authors should more clearly articulate the core contribution of this work.

2.The authors discuss a large number of related works in Table 1; however, their experiments do not include comparisons with the most recent methods (e.g., EDP-GNN and GDSS), which raises concerns about the validity of their experimental results.

**Questions:**

Please refer to weaknesses

---

> ### Author Response · Authors · 2025-11-20
>
> 1- “Upsampling appears to be a popular approach in graph generation (please correct me if I'm mistaken). The authors should more clearly articulate the core contribution of this work.”
> -- If reviewer means upscaling with “Upsampling”, unfortunately, it is not a popular approach due to various reasons, not just because no one has considered. VAE-style models (like GraphVAE) decode a fixed adjacency matrix. So the decoder architecture itself hard-codes the max graph size, so you can’t just ask for a 10× larger graph at inference. Autoregressive models (GraphRNN, GRAN, etc.) generate a sequence over nodes and edges. Therefore, again trained on a particular size range and cannot upscale. There are some approaches that can be counted as an upscaling approach, such as Barabási–Albert that uses that use preferential-attachment-based mechanism, or probabilistic-based base models, but they are the models that do not learn directly from the data. Therefore, we deeply believe our GraphK model represents an important step toward permutation-invariant graph generation, with strong time and complexity efficiency compared to SOTAs and, importantly, the ability to upscale to thousands of nodes.
>
> 2- “The authors discuss a large number of related works in Table 1; however, their experiments do not include comparisons with the most recent methods (e.g., EDP-GNN and GDSS), which raises concerns about the validity of their experimental results.”
> Thank you for the suggestion. We are aware of both models and included them in our literature review (lines 138–139). However, they are not suitable baselines for our comparison. EDP-GNN is an old model (2020) designed for very small graphs. As reported in its own paper (Niu, 2020) and in Kong et al. (2023), it does not outperform models such as GRAN or GraphVAE on larger graphs, and it is not intended for graphs with hundreds or thousands of nodes. GDSS is another diffusion-based model, but DiGress has been shown to outperform GDSS on multiple datasets. Moreover, GDSS again focuses on small-scale(up to 400 nodes) graph generation and is not suitable for graphs with more than 1k nodes, as also seen in their claims and evaluation results. Our goal was to choose baselines that best highlight our contributions in scalability, time efficiency, permutation invariance, and generation of large graphs, areas that most recent graph generative models do not address.

---

### Official Review · Reviewer_hVeq · 2025-11-05

**Soundness:** 2
**Presentation:** 2
**Contribution:** 2
**Rating:** 2
**Confidence:** 4

**Summary:**

This paper proposes a methods, named GraphK, which is encoder-sampler-decoder framework for graph generation that overcomes the challenges of scalability, flexibility in generating the number of nodes (upscaling /downscaling) and ability to model underlying structures. GraphK learns permutation-invariant latent representations and sampling new node embeddings via maximum likelihood estimation and predicts edge by using KDTree based top-k neighbor search in latent space.

**Strengths:**

The authors focuses on an important problem of  scalability for graph generation.

**Weaknesses:**

1. Over all the paper is hard to follow with unclear details for example, the abstract mentions that GraphK provides fine-grained control to generate graphs , however there is no mention of what controls are?  Is the paper referring to spectral, orbit, and motif mention in the result Table 2? If so, Is spectral, orbit, and motif are the only attribute chosen? Was there any motivation of using only those ?
2. As evident from Figure 2, other baseline models performs better than proposed approach GraphK. Number of nodes and edges predicted by GraphK are more far away from the ground truth/original graph compared to the other baseline models there by questioning the effectiveness of the approach.
3. The paper would benefit by comparing from more recent baselines for example like GruM, EDGE, and SPECTRE
4. Over all, the evaluation is weak and paper would benefit with more clear explanation of fine-grained control.

**Questions:**

see above

---

> ### Author Response · Authors · 2025-11-20
>
> 1- “Over all the paper is hard to follow with unclear details for example, the abstract mentions that GraphK provides fine-grained control to generate graphs , however there is no mention of what controls are?”
> -- We agree with reviewers that “fine-grained control” might be prone to misunderstanding. We meant upscaling and downscaling ability, since we can arrange the scale (how much we want to upscale or downscale). And we just used this wording in the abstract with this sentence: “GraphK enables both upscaling (generating graphs with more nodes than the input) and downscaling, providing fine-grained control over output graph size.” We updated this it in the rebuttal version.
>
> 2- “Is the paper referring to spectral, orbit, and motif mention in the result Table 2? If so, Is spectral, orbit, and motif are the only attribute chosen? Was there any motivation of using only those ?”
> -- No. They are not related to the written “fine-grained” intention. As it is written between lines 370-372, Maximum Mean Discrepancy (MMD) is a kernel-based distance between probability distributions, and spectral distances, orbit counts, and motif distributions come from the choice of kernel. The motivation that we are using these kernels and MMD is that all other SOTAs are using them, as it is also referenced in the paper, and giving a fair comparison.
>
> 3- “The paper would benefit by comparing with more recent baselines, for example, like GruM, EDGE, and SPECTRE”.
> -- Thank you for the suggestion. We would like to include more diffusion-based baselines, but existing diffusion models are generally not suitable for the large graphs we target (e.g., social networks or community graphs with 1k+ nodes) due to their O(TN^2) time and memory complexity, as discussed in our paper. For this reason, we only include DiGress (Vignac et al., 2022) from diffusion-based family, since it is the SOTA diffusion model that scales beyond small molecular graphs. Regarding SPECTRE, the DiGress paper reports that DiGress outperforms SPECTRE across multiple datasets and metrics, so we did not include SPECTRE in our evaluation. Moreover, we were unable to find a model named GruM despite our extensive searching. Could the reviewers provide a reference and clarify why they believe GruM is particularly relevant? Overall, we tried to select baselines that best highlight our contributions in scalability, time efficiency, permutation invariance, and generation of large graphs, areas that most recent graph generative models do not address.
>
> But instead, we included two more baselines to our comparison (PARD[1], BiGG[2]) and included the results in the rebuttal version. Our results show that GraphK outperforms (and competitive on some metrics) PARD and BiGG .
>
> [1] Zhao, Lingxiao, Xueying Ding, and Leman Akoglu. "Pard: Permutation-invariant autoregressive diffusion for graph generation." Advances in Neural Information Processing Systems 37 (2024): 7156-7184
> [2] Dai, Hanjun, et al. "Scalable deep generative modeling for sparse graphs." International conference on machine learning. PMLR, 2020.

---

### Author Response · Authors · 2025-12-03
**Summary of Changes Made**

We revised the paper based on the reviewers' comments and made several changes.

(1)- Several reviewers asked for more baselines. In response, we added two additional models, PARD[1] and BiGG[2], and included their results in the revised version. We chose these two baselines for the following reasons:

(1a)- BiGG is one of the few existing models that can handle large graphs and is widely considered the SOTA for large graph generation. Since our method focuses on scaling to graphs with thousands of nodes, BiGG is the only practical and relevant baseline at this scale.

(1b)- PARD is the most recent permutation-invariant baseline. None of the previous baselines we used was truly permutation invariant. By including PARD, we show how a permutation-invariant approach behaves in this setting and how GraphK compares to the newest model in this category.

(1c)- The new large-graph experiments support this choice. GraphK generates graphs with 10k–50k nodes in under 10 seconds, which is far faster than BiGG. For comparison, BiGG reports about 7 minutes for 10k nodes and roughly 20 minutes for 50k nodes. Including PARD also shows that a permutation-invariant model can scale, since PARD itself cannot generate even the CiteSeer graph with 2k nodes. Therefore, with these new experiments, we show that a permutation-invariant model can scale and still stay competitive with SOTA methods for large graphs. Overall, GraphK’s main contribution is showing that using k-neighborhoods and a GMM-based sampler makes it possible to build a model that is both scalable and permutation invariant. Even though edges are generated independently, GraphK still captures key graph patterns like sparsity and power-law degrees, showing that these choices do not harm realism.

[1] Zhao, Lingxiao, Xueying Ding, and Leman Akoglu. "Pard: Permutation-invariant autoregressive diffusion for graph generation." Advances in Neural Information Processing Systems 37, NeurIPS, (2024): 7156-7184

[2] Dai, Hanjun, et al. "Scalable deep generative modeling for sparse graphs." International conference on machine learning. PMLR, (2020).

(2)- We added an encoder ablation by including a third encoder, HOPE, to show that the framework works with different embedding methods.

(3)- We updated the abstract to resolve the confusion around "fine-grained control" and performed additional proofreading.

****
For the reviewers' discussion and engagement:

(1)- We explained why some suggested models are not suitable baselines (this was already written in the paper, so no update was needed). EDP-GNN, GDSS, SPECTRE, and most diffusion-based methods are designed for small graphs and cannot scale to thousands of nodes, which is the setting of our work. We also mentioned that we could not find a model named "GruM", as suggested by the reviewer.

(2)- To better communicate our main contribution, we discussed why upscaling is not common in existing graph generators, cannot upscale to larger sizes and how GraphK addresses this limitation. VAE-style models and autoregressive models cannot change graph size at inference, and the paper already explains how GraphK avoids these limits, so no change was needed.

(3)- In response to questions about permutation invariance, we discussed how the GMM sampler captures the global distribution of node embeddings in a way that does not depend on node order, and why the encoder itself does not need to be permutation invariant. These explanations were already in the main manuscript (now moved to Appendix 5 to make space for the new scalability experiments with BiGG). Appendices 9 and 10 also existed in the initial version, which discuss more on permutation-invariant evaluation, so no change was made.

(4)- We also answered that our method works best on graphs with irregular structures and is less effective on highly regular graphs such as grids or lines. This reflects a common dilemma in graph generation, where no single model performs best across all graph types or datasets, and we had already stated this in the limitations section, so no update was needed.

---

### Note · Program_Chairs · 2026-01-17
**Submission Desk Rejected by Program Chairs**

The following references in this submission do not refer to real documents and/or have major errors in bibliographic information:

 Junchi Xu, Pan Li, Chuan Zhang, and Stefanie Jegelka. Learning random graph models with graph neural networks. In Proceedings of the Twenty-Ninth International Joint Conference on Artificial Intelligence (IJCAI), pp. 3954-3960, 2020.